# Curing the Curable: Managing Low-Risk Acute Lymphoblastic Leukemia in Resource Limited Countries

**DOI:** 10.3390/jcm10204728

**Published:** 2021-10-15

**Authors:** Bernice L. Z. Oh, Shawn H. R. Lee, Allen E. J. Yeoh

**Affiliations:** 1VIVA-University Children’s Cancer Centre, Khoo-Teck Puat-National University Children’s Medical Institute, National University Hospital, Singapore 119074, Singapore; paeolzb@nus.edu.sg (B.L.Z.O.); paelhrs@nus.edu.sg (S.H.R.L.); 2Department of Paediatrics, Yong Loo Lin School of Medicine, National University Singapore, Singapore 119074, Singapore

**Keywords:** childhood acute lymphoblastic leukemia, low-risk ALL, risk-stratified treatment, treatment related toxicity

## Abstract

Although childhood acute lymphoblastic leukemia (ALL) is curable, global disparities in treatment outcomes remain. To reduce these global disparities in low-middle income countries (LMIC), a paradigm shift is needed: start with curing low-risk ALL. Low-risk ALL, which accounts for >50% of patients, can be cured with low-toxicity therapies already defined by collaborative studies. We reviewed the components of these low-toxicity regimens in recent clinical trials for low-risk ALL and suggest how they can be adopted in LMIC. In treating childhood ALL, the key is risk stratification, which can be resource stratified. NCI standard-risk criteria (age 1–10 years, WBC < 50,000/uL) is simple yet highly effective. Other favorable features such as *ETV6-RUNX1*, hyperdiploidy, early peripheral blood and bone marrow responses, and simplified flow MRD at the end of induction can be added depending on resources. With limited supportive care in LMIC, more critical than relapse is treatment-related morbidity and mortality. Less intensive induction allows early marrow recovery, reducing the need for intensive supportive care. Other key elements in low-toxicity protocol designs include: induction steroid type; high-dose versus low-dose escalating methotrexate; judicious use of anthracyclines; and steroid pulses during maintenance. In summary, the first effective step in curing ALL in LMIC is to focus on curing low-risk ALL with less intensive therapy and less toxicity.

## 1. Introduction

Childhood acute lymphoblastic leukemia (ALL) is curable. Underpinning the cure for ALL is more than half a century of intensive collaborative research [1] that has systematically tested and defined highly effective drug combinations which now form the backbone of contemporary protocols. Although there are minor differences, contemporary ALL protocols are strikingly similar and almost formulaic. However, as the survival of childhood ALL in high-income countries (HIC) surpasses 90% [1], significant disparities in survival have emerged [2]. The high cure rates of ALL achieved in HIC are not seen in low-middle income countries (LMIC) [3]. With 80% of the world childhood ALL burden residing in LMIC [4,5], our success in curing childhood ALL remains limited and geographically restricted [2]. To reduce such glaring disparities, many groups such as the International Pediatric Oncology Society (SIOP), VIVA Foundation for Childhood Cancer, St Jude Global, and the World Health Organization are beginning to tackle the obstacles to widespread adoption of effective treatment. The first steps in improving cures for ALL worldwide, we believe, is for LMIC to focus on curing low-risk ALL as it is highly cost-effective and transformational. In this review, we focus on key components of contemporary trials on curing low-risk ALL, cost-effectively.

## 2. Causes of Failures in LMIC

We reference the World Bank Income Group classification in defining LMIC (https://datahelpdesk.worldbank.org/knowledgebase/articles/906519-world-bank-country-and-lending-groups accessed on 1 October 2021). Specifically, in 2021, LMIC are defined as countries with gross national income per capita of <USD 12,695. Albeit imperfect and simplistic, this definition is used in the Lancet Oncology Commission on Sustainable Care for Children with Cancer [5], in which we also participated.

In HIC, the main cause of failure in treating childhood ALL is relapse [6]. This fear of relapse is so ingrained that overtreatment is rarely questioned. However, for LMIC, the key reason for failure is treatment toxicity [7]. Treatment toxicity disproportionately inflicts suffering and exponentially increases the cost of treatment [5], which then invariably leads to treatment abandonment [8]. In LMIC, treatment toxicity is further amplified by malnutrition [9], suboptimal supportive care [10], and widespread antibiotic resistance [11]. The Malaysia-Singapore ALL study group (Ma-Spore) is based in Malaysia and Singapore, an example of two countries which have emerged from LMIC status within the past three decades [12]. Given the health resource constraints, the Ma-Spore study group focused on testing cost-effective deintensification of therapy in low-risk ALL patients as one of the main strategies.

## 3. Identifying Low-Risk Groups in Resource Limited Settings

The key to better cures in ALL is better risk stratification [13]. Specifically, if we can define each patient’s risk of relapse early, the optimal intensity of therapy can be given to maximize cure while minimizing side-effects. The two major determinants of relapse are molecular genetics and early response to therapy [6]. Early response to therapy can be measured by (1) widely available peripheral blast counts by light microscopy on day 8; and/or (2) sophisticated minimal residual disease (MRD) quantitation of sub-microscopic disease by flow cytometry or polymerase chain reaction (PCR) [2]. Based on the resources of specific hospitals and the country level of care for ALL and access to tests, we propose various levels for risk assessment of ALL (Table 1).

In LIC with very limited resources, the only diagnostic facility available may be light microscopy to identify ALL lymphoblasts. Flow cytometry may not be available to subtype ALL into B or T-ALL. Given these limitations, a simple chest X-ray revealing a mediastinal mass [28] may help in differentiating T-ALL from B-ALL. In these resource limited settings, early responses to treatment can be assessed morphologically using day 8 peripheral blast count (>1 × 10^9^/L) and day 15 or end-of-induction (EOI) bone marrow morphology [2]. ALL may have to be managed by general pediatricians without access to specialized pediatric oncology nursing care in LIC [10]. Chemotherapy drug availability is also likely to be limited and its supply unreliable [29,30]. Because of these constraints in LIC, it is best to have one simple, common protocol that is minimally myelosuppressive. The SIOP PODC [31] and the Lancet Oncology Asian Consensus Protocol [2] are probably effective stratified regimens.

For MIC, limited panel flow cytometry to diagnose B- and T-ALL may be possible. Working with universities, MIC hospitals can potentially access PCR thermocyclers [12]. With PCR thermocyclers it is possible to run simple oncogene fusion tests to screen for *BCR-ABL1* and *ETV6-RUNX1* fusions [32,33]. In these settings, it is important to put in appropriate positive and negative controls as cross contamination and degraded RNA are common.

The ALL IC-BFM 2002 study group demonstrated the feasibility of a risk stratification approach based on a combination of modified NCI criteria, early morphologic evaluation on days 8, 15, and 33, without PCR-MRD monitoring. The study was successfully implemented in 15 countries across 3 continents in 130 centers. The ALL IC-BFM 2002 [16] study reported an excellent 81% EFS and 90% OS in the standard risk arm. Interestingly, the ALL IC-BFM 2002 standard risk criteria were defined as age 1 to 6 years and lower white blood cell (WBC) count of 20 × 10^9^/L. The ALL IC-BFM 2002 protocol was intensive, and treatment administered in national centers with good supportive care.

## 4. The Importance of the NCI Standard-Risk (SR) Criteria

ALL is a genetically heterogeneous disease [13,34]. The National Cancer Institute (NCI) standard-risk (SR) criteria, presenting age of 1 to 10 years and WBC count < 50 × 10^9^/L), are simple yet surprisingly effective risk stratification criteria [35]. Favorable genetic drivers, such as hyperdiploidy and ETV6-RUNX1, form the largest proportion of NCI SR patients [34]. In MS2003 [18], age remained prognostically significant for event-free survival (Figure 1). The NCI criteria can be easily applied even in LMIC settings and should remain one of the mainstays of risk stratification [2].

Table 2 summarizes the risk stratification criteria used by various clinical trials to characterize patients with ALL who are at the lowest risk of relapse. Overall, the NCI SR criteria remain a cornerstone of ALL risk stratification for 9 of the 13 clinical trials evaluated in this review. However, the upper age limits for inclusion in these low-risk arms still vary from group to group (Table 2).

Like those of the AIEOP-BFM ALL 2000, the NCI features were not used for risk stratification in MS2003 [18]. However, compared to NCI SR patients, NCI HR patients treated in the MS2003-SR arm did significantly more poorly (Figure 1). Given these findings, in the MS2010 [19], patients aged ≥10 years old were treated in either the intermediate or high-risk arm depending on MRD responses—and not in the lowest risk, SR arm.

In the AIEOP-BFM ALL 2000-SR arm [20], EOI MRD-negative NCI HR patients had poorer outcomes on the less intensive Protocol III compared to standard Protocol II during delayed intensification (DI) (8-y EFS: 82.9% versus 90.4% *p* = 0.04). In contrast, NCI SR patients did equally well. The AIEOP-BFM ALL 2000 SR study concluded that, despite negative EOI MRD, age >10 years adversely affected outcomes. Taken together, despite EOI MRD negativity, both the MS2003 and ALL AIEOP-BFM ALL 2000 studies suggested that NCI HR patients, specifically teenagers, should *not* be treated with de-intensified treatment.

## 5. Favorable ALL Genetics: Hyperdiploidy and ETV6-RUNX1

Hyperdiploidy (>50 chromosomes) and *ETV6-RUNX1* ALL are associated with excellent outcomes (5-year EFS >90%). Unfortunately, the karyotyping of lymphoblasts to determine ploidy is technically challenging and different from antenatal karyotyping. Hyperdiploidy is characterized by recurrent, non-random gains in specific chromosomes: 4, 10, 17, and 18. To standardize the diagnosis of hyperdiploid ALL, the Children’s Oncology Group (COG) focused on double or triple trisomy fluorescent-in-situ-hybridization (FISH) probes for chromosomes 4, 10, and/or 17 to define these favorable hyperdiploid features.

*ETV6-RUNX1* is an oncogene fusion transcript and cannot be defined by conventional karyotyping. UKALL’s strategy of low-cost FISH to identify oncogene fusions such as *ETV6-RUNX1*, *BCR-ABL1* fusion probes and a *KMT2A* break-apart probe in a triple probe FISH screening strategy has been tested in low-resource settings [38,39]. In contrast, Ma-Spore and other groups have used reverse-transcription PCR (RT-PCR) to screen for *ETV6-RUNX1*, *BCR-ABL1, TCF3-PBX1,* and *KMT2A-AFF1 (AF4).* RT-PCR can be performed using standard PCR thermocyclers, which are also available in many universities including those in LMIC. Although more expensive, there are also available oncogene fusion screening kits for leukemia, e.g., QuanDx’s Leukemia Fusion Gene (Q30) Screening Kit and the HemaVision Screen kit. Given that the instability of mRNA and that PCR reactions may be prone to aerosol contamination, proper positive and negative controls are critical for RT-PCR screening. Referencing and partnering with good laboratories internationally, such as VIVA-NUS CenTRAL and the St Jude Global Alliance, can also be very helpful.

Using a low-intensity protocol, COG AALL0331 [23] demonstrated excellent outcomes in children who achieved EOI MRD negativity with either favorable triple trisomy (38% of population) of chromosomes 4, 10, 17, or ETV6-RUNX1 (62% of population). COG AALL0331 showed that intensification of therapy for these low-risk patients did not improve outcomes. For this low-risk cohort, the successor COG AALL0932 [22] also reported excellent outcomes: 5-year DFS 98.8% and 5-year OS 100% with a P9904-based regimen without alkylating agents or anthracyclines in the LR-M arm. However, only 6.5% of the AALL0932 study population was eligible for this randomization; and only 3.3% of patients were assigned to the P9904-based LR-M arm. Critically, to be able to define this lowest-risk subgroup, there is a requirement for excellent cytogenetics or FISH setup defined hyperdiploidy, oncogene fusion screening for ETV6-RUNX1, and EOI MRD. For LMIC, accurate EOI MRD by flow may not be available.

In the presence of hyperdiploidy and ETV6-RUNX1, poor responses such as high EOI MRD of >1% are fortunately very rare. In MS 2003/2010 and UKALL 2003, only 3% of patients with low-risk genetics had a high EOI MRD of >1% [40]. Hence MRD monitoring for this low-risk genetic group is probably not critical. For MIC, using DNA index >1.16 may be feasible, as demonstrated by the RELLA05 [14] group in low-resource settings in Brazil. FISH for double trisomy 4 and 10 as a surrogate marker for triple trisomy is also feasible.

## 6. Democratization of Flow Cytometry

Interestingly, it is common for hospitals, even in LMIC, to have a good flow cytometer. The widespread availability of good, multi-color flow cytometers makes it possible to do flow cytometry for diagnosis of ALL. However, not many laboratories are trained to properly perform flow cytometry for the diagnosis of childhood leukemias. Access to a supply chain of good quality fluorochrome-labeled antibodies is also potentially a problem.

While several simple low-cost flow cytometry methodologies to measure MRD have been developed [41,42], they have yet to be widely implemented in Asia. Flow MRD needs to be analyzed and interpreted properly. In the presence of a lot of hematogones, simple, low-cost flow MRD using a limited panel of markers can yield misleading results. Flow MRD-lite end-of-induction assessment has been used in limited resource settings such as the RELLA05 study [43]. Although MRD testing is expensive, its key role in risk assignment would offset costs involved in toxicity related hospitalizations. In addition, it can be cost effective to set up a good flow MRD-lite platform to identify the best responders that can be cured with less therapy.

## 7. Specific Considerations for T-Lineage ALL

Treatment de-intensification in T-ALL is much less studied and should be undertaken with caution in the context of a clinical trial. Outcomes in T-ALL have only very recently improved significantly, approaching those of B-ALL. This has been achieved with combinations of (1) the use of dexamethasone (Dexa)-based 4 drug induction, (2) a more intensive Berlin-Frankfurt-Munster (BFM) ALL backbone, (3) Capizzi escalating methotrexate, and (4) optimizing the use of L-asparaginase (L-asp). In the MS2003 study [18], which was Dexa-based, 6-year EFS rates of B and T-ALL patients were 80.7% and 80.5%, respectively.

Use of Dexa throughout all phases of therapy, like in the MS2003 [18] and UKALL 2003 [27] studies, has led to better outcomes in children with T-ALL. This better outcome with Dexa could be due to better CNS penetration, given that CNS relapse is more common in T-ALL. T-ALL patients should receive 4-drug Dexa-based induction but will require prolonged inpatient admission throughout the whole period of induction because of high risk of infections and TRM.

The largest T-ALL study, the COG AALL0434 study [44], surprisingly showed superior outcomes with the Capizzi escalating MTX plus L-asp regimen compared to HDMTX regimen (4-y DFS 92.5% ± 1.8% vs. 86.1% ± 2.4%, *p* = 0.02). However, 90% of T-ALL patients on AALL0434 received cranial radiotherapy. AALL0434 also found that addition of 5 days of nelarabine improved outcomes for IR and HR T-ALL (4-y DFS 88.9% vs. 83.3%). However, the high cost and high neurotoxicity of nelarabine will limit its use in LMIC.

## 8. Delaying the First Intra-Thecal (IT) Chemotherapy

Traumatic lumbar puncture (LP) with blasts is a risk factor for CNS relapse. If the first LP is performed at the time of diagnosis [45], traumatic LP occurs in up to 14%. Delaying the first IT until after clearance of circulating blasts at the end of the first week of induction would reduce incidence of traumatic LP with blasts, an adverse risk feature. This was first described in the Tokyo Children’s Cancer Study Group study L89-12 [46]. The TPOG-ALL-2002 study [47] also confirmed that the delay of the first triple IT did not adversely affect survival or CNS control despite omission of cranial irradiation.

Delaying of the first IT also reduces the risk of methotrexate-related kidney injury that may be exacerbated by ongoing tumor lysis syndrome during the induction phase. Reduced need for sedation in the first week of therapy may also be advantageous to patients with large mediastinal masses at diagnosis, given the inherent risks of airway obstruction with procedures requiring sedation in such cases.

## 9. Prednisolone/Dexamethasone-Based and 3/4-Drug-Based Induction

Pred has historically been used during the BFM ALL induction protocol, while Dexa has been used later during DI (Figure 2). Dexa is more potent than Pred and because of better CNS penetration [48], reduces the rate of CNS relapse. The enduring question of whether Dexa is superior to Pred during induction was tested in the randomized AIEOP-BFM ALL 2000 [36] study in a 4-drug induction including anthracyclines. AIEOP-BFM ALL 2000 showed that patients with a good Pred response who received Dexa during induction had one-third the risk of relapse of those who had received Pred—a remarkable feat. However, these improvements in relapse rates were offset by the higher incidence of life threatening events during induction. Overall, despite the marked reduction in relapse rates in the Dexa arm, there were no differences in OS as relapses in the Dexa arm were less salvageable and more patients died of infections during Dexa-based induction. Subsequently the AIEOP-BFM group reserved 4-drug Dexa-based induction only for a subset of T-ALL patients with good Pred response.

The Japanese L95-14 [49] and the Dana-Farber Cancer Institute ALL 91-01P [50] trials also reported a higher rate of infection-related induction deaths in the Dexa arm as compared to those who had received Pred. Infectious deaths also increased during Dexa-based induction in UKALL 97 [51] although there was overall survival benefit.

To use Dexa during induction, a 3-drug induction without anthracyclines is feasible in LMIC. Of note, many groups such as the COG, UKALL, and the Ma-Spore ALL used Dexa-based, 3-drug induction without anthracyclines. COG and UKALL used it for NCI SR induction while Ma-Spore used it for all B-ALL patients. Without anthracyclines, Dexa-based 3-drug induction can be given safely and mainly as outpatient therapy.

Dexa-based 4-drug induction is best reserved for T-ALL patients and used in hospitals with good isolation facilities and ability to treat breakthrough secondary infections including fungal infections. These hospitals should also have good microbiological diagnosis platforms for bacterial, fungal, and viral pathogens. Dexa-based 4-drug induction is toxic, even in the context of HIC where supportive care is good. We do not recommend Dexa-based 4-drug induction for LMIC.

## 10. L-Asp Doses in Induction and Delayed Intensification

A key drug in treatment of ALL is L-asp. Unlike adult ALL protocols, such as Hyper-CVAD, pediatric-inspired protocols use L-asp as a mainstay drug during induction and DI. L-asp is an enzyme derived from *E. coli* that can cause allergic reactions and silent inactivation. L-asp is moderately myelosuppressive and can cause pancreatitis and thromboembolism, especially in children > 10 years old.

After the first exposure to L-asp during induction therapy for newly diagnosed ALL, the risk of neutralizing antibodies and silent inactivation is low. Because of the lower risk of neutralizing antibodies that have to be overcome with higher doses, Ma-Spore ALL induction starts with a lower dose of L-asp during induction (7500 U/m^2^ of Leunase spaced out to twice a week). This lower dose of L-asp during induction also reduces the risk of myelosuppression. During DI, where low levels neutralizing antibodies may already have developed, we use a higher dose of L-asp of 10,000 U/m^2^ every 3 days. This high dose allows for sufficient asparagine depletion during DI.

The major brands of L-asp available include: Leunase (Kyowa-Hakko), Kidrolase (Kyowa-Hakko), and Spectrila (Medac, Germany). In addition, L-asp is also manufactured by a few companies in India. The various brands of L-asp have different potencies and different risks of allergic reaction. Pegylated (PEG) L-asp has a much longer half-life than regular L-asp. However, PEG-L-asp is expensive and not registered in most LMICs. Because of these limitations, the Ma-Spore ALL study group focused on using L-asp. Erwinase is given to patients with allergic reactions to L-asp and PEG L-asp. However, Erwinase is less potent and has a much shorter half-life requiring dosing of 20,000 U/m^2^ every 2 days to ensure complete asparagine depletion.

The St Jude Total XVI [26] study showed prolonged and more intensive asparagine depletion using higher doses of PEG L-asp (3500 U/m^2^ versus 2500 U/m^2^) did not improve outcomes. Instead, it was associated with a higher incidence of toxic deaths than in an earlier study (3.2% vs. 1.4%). This prolonged asparagine depletion is also associated with increased risk of pancreatitis and long-term poor pancreatic function with diabetes mellitus.

In MS2010 [52], single doses of vincristine and L-asp were added during DI to maintain treatment intensity during a rest period at day 15. However, this led to more hospitalizations for fever, increased risk of bacteremia, and critical-care admissions, but fortunately without any increase in treatment-related mortality. The DFCI-ALL 05-01 [53] also previously described the myelosuppressive effects of asparaginase.

Because of its high costs, risks of allergy and silent inactivation, we recommend restricting L-asp use to only the induction and DI phases, especially in LMICs. To reduce the risk of allergy, Ma-Spore delayed L-asp until after at least 2 days of steroid cover had been started. We also caution against any additional doses of L-asp given that it causes increased myelosuppression and a higher risk of infections.

## 11. Anthracycline-Free Regimens

Anthracyclines are most used as part of induction and DI in the BFM-ALL treatment backbone. Although effective, anthracyclines cause severe immediate myelosuppression and long-term cardiotoxicity [54,55]. Because of these side effects, Ma-Spore and other groups have attempted to eliminate or reduce anthracyclines in the treatment of low-risk ALL. The COG AALL0932 [22] and MS2010 are both examples of clinical studies where anthracyclines were completely omitted in their low-risk arms. In terms of toxicity, MS2010-SR [52] revealed excellent results comparable to those of other contemporary protocols, yet with reduced toxicity. As mentioned above, in a highly selected subgroup with low-risk genetics and that were EOI MRD negative, the COG AALL 0932 LR-M arm reported excellent results with 5-year DFS of 98.8% and 5-year OS of 100% with no anthracyclines and alkylating agents.

The CCG-105 [56] study showed that dose dense DI is only critical for *older* children. This is due to residual leukemia cells that persist after induction/consolidation which are relatively resistant to therapy. Anthracyclines, usually doxorubicin, are used with Dexa, vincristine and L-asp (Protocol II) for intensive DI. However, for younger children who have low-risk ALL, the CCG-105 study showed that dose intensive DI is probably not critical.

## 12. Is High-Dose Methotrexate Really Necessary?

In LMIC, it is difficult to administer high-dose methotrexate (HD MTX) safely. This is because IV MTX > 500 mg/m^2^ requires folinic rescue dosing and close MTX level monitoring. Although the adjustment of the start time for administration can allow MTX level monitoring during office hours, this infrequently utilized test is generally not available and not cost-effective in most LMIC settings. Although various groups have devised various strategies [57,58] to overcome challenges of giving HDMTX, we review alternatives to HD-MTX in Table 3.

For all NCI-SR patients, instead of HDMTX, the COG study used escalating intravenous MTX that started at 100 mg/m^2^ and did not require serum MTX level monitoring. COG AALL0331 [23,37] and COG AALL0932 [22] reported excellent outcomes in NCI SR patients treated without HD MTX. The UKALL 2003 [27] achieved excellent outcomes without use of HDMTX for all patients. Specifically, for EOI MRD negative patients (Regimen A and B), there was no HDMTX or Capizzi MTX-L-asp; while EOI MRD positive patients had two blocks of Capizzi MTX-L-asp.

Although the COG AALL0232 [59] showed that HDMTX was superior to Capizzi MTX, the reported benefits of HDMTX over Capizzi MTX were in fact in *higher risk* B-ALL but not low-risk ALL. The newer UKALL 2011 randomized EOI MRD negative patients to receive HDMTX compared to interim maintenance.

## 13. Delayed Intensification—Is More Necessarily Better?

Although the importance of DI is clear, a balance between dose intensity and treatment toxicity is paramount. The BFM/COG DI Protocol II is intensive with significant toxicity, thus many groups have focused on deintensification of DI. In SR patients who were MRD negative, DCOG-ALL10 [24] successfully removed doxorubicin, cyclophosphamide, cytarabine, and thioguanine, which were replaced with a single, low-intensity Protocol IV, which consists of Dexamethasone, two doses of Vincristine, and single doses of PEG L-asp and intrathecal chemotherapy; with excellent outcomes (93% 5-y EFS and 99% 5-y OS). The randomized CoALL 07-03 study [21] also successfully de-intensified Protocol II, by removing one dose of doxorubicin and one week of Dexa in SR patients. In contrast, the large, randomized AIEOP-BFM-ALL 2000 study [20] showed that the shortened but dose-dense Protocol III was paradoxically more toxic and less effective in preventing relapse. The AIEOP-BFM ALL Protocol III is shorter and highly compressed DI, resulting in more toxicity and prolonged post-Protocol III delay.

Recent studies have demonstrated that repeated DI blocks might not improve outcomes. Figure 3 summarizes the various strategies and overviews of major clinical trials in childhood ALL. The randomized ALL IC-BFM 2002 study [16] failed to show any improvement in outcomes with additional DI blocks in both standard and medium-risk patients. Similarly, CCG-1991 [60] showed no added benefit with double DI blocks in patients with standard-risk ALL. Instead, escalating MTX during interim maintenance improved outcomes. UKALL2003 randomized EOI MRD negative patients to single versus two blocks of Protocol II; one block of Protocol II was less toxic without compromising outcomes [27].

In the MS2003 study [18] SR arm, the DI phase consisted of two blocks of Protocol III like the experimental arms of the ALL IC-BFM 2002 study [16]. Unfortunately, as in the ALL IC-BFM 2002, there was significant toxicity during DI in MS2003 where most of the treatment-related deaths occurred [52]. Although larger randomized clinical trials investigating the effects of DI indicate that a single block of BFM/COG Protocol II is probably sufficient, the Ma-Spore chose to continue with two shorter but further modified DI blocks instead of a single block of Protocol II. In the MS2010, EOI MRD negative patients received two less intensive DI blocks without anthracyclines (Protocol V) with less treatment interruption and toxicity. Toxicity analysis of MS2010 [52] revealed significant reductions in toxicity in terms of infections as well as overall phase delays.

For low-risk ALL, it is not clear whether a strong intensive DI phase is necessary. Taken together, one block of COG Protocol II DI has been shown to be highly effective and is our recommendation for LMIC with good supportive care.

## 14. The Malaysia Singapore Experience

The Ma-Spore Study Group is a collaborative group of four pediatric oncology units from Malaysia and Singapore. Ma-Spore started with a MRD risk-stratified, Ma-Spore ALL 2003 (MS2003) treatment protocol [12,18]. Because of moderate resources, MS2003 focused on deintensifying therapy in MRD-negative patients.

MS2003 starts with a less myelosuppressive 3-drug Dexa-based induction to reduce the risk of severe infections during induction. The Ma-Spore treatment mantra is *“Patient first, leukemia second.”* The aim was to get the patient to safety first by allowing recovery of marrow function. Depending on MRD response after induction, strength of delayed intensification therapy is tailored later to eliminate residual leukemia. The vast majority of patients with no high-risk genetics and a good day 8 Pred response received 3-drug Dexa-based induction without anthracyclines. In Ma-Spore ALL studies, MRD risk stratification is by using a single PCR MRD marker at EOI and at the end of consolidation (EOC) at week 12. MS2003 focused on intensive DI by adopting the experimental ALL IC-BFM 2002 repeated Protocol III blocks. MS2003 [12] achieved a 6-year EFS of 80%, with an overall survival of 88%.

In addition, a strong collaborative network was forged between the two countries, where bone marrow MRD samples were processed and cryopreserved then couriered weekly on dry ice to the centralized laboratory in Singapore. There was also regular exchange of manpower training and knowledge sharing between the different pediatric oncology centers that extended to regional hospitals in rural areas in Malaysia. An important feature was regular telephone calls to regional hospitals to track count recovery on full blood counts, drug doses, and complications. Regional hospitals were educated on complications such as febrile neutropenia, to be able to reach emergency services in a timely manner should the need arise. Healthcare personnel in rural areas were also educated on the management of neutropenic fever and the importance of up triaging and early administration of antibiotics.

The keys to the success of Ma-Spore are the use of centralized academic molecular laboratories in the National University of Singapore and University Malaya, and a protocol design that is cognizant of moderate access to supportive care. Figure 4 summarizes key networks that have also been used in the Malaysia Singapore experience in establishing a program to treat children with ALL in settings with resource limitations. 

## 15. Infections

In ALL, given the growing threat of antimicrobial resistance and, more recently, the COVID-19 pandemic, treatment-related infections are a major concern. The risk of infection during ALL treatment is dependent on the treatment phase and its intensity. The induction phase poses the highest risk of infections due to a combination of prolonged myelosuppression from both disease and induction chemotherapy [61,62]. Because of this exquisite vulnerability to sepsis during induction, the Ma-Spore group focused on 3-drug Dexa-based induction, which is less myelosuppressive, yet sufficiently intense to achieve sufficient complete remission to promote marrow recovery.

Although UKALL 2003 [27] had no HDMTX blocks, which made it feasible in limited resource settings, the risk of sepsis during the 4-drug Dexa-based induction protocol was still significant [62]. Given the septic deaths prevalent even in high income settings, these risks may be exponentially higher in limited resource settings. In areas with hygiene concerns and where access to supportive care may be an issue, it may be prudent to keep patients within a closer proximity during such high-risk periods.

In the St Jude Total XV [61] study, the lack of neutrophil surge after Dexa pulse, as a reflection of decreased marrow reserves, was linked to a high risk of sepsis. Dexa suppresses fever. During Dexa pulse, presence of even low-grade fever of >37.3 °C and severe neutropenia (ANC < 0.5 × 10^9^/L) confer increased risk of sepsis. In limited resource settings, neutropenic fever during Dexa-based phases should be prioritized for emergency access to supportive care.

Malnutrition aggravates the risk of infection during cancer treatment [9]. In limited resource settings, training shared-care hospitals and educating families to recognize fever and signs of sepsis is critical. This involves providing clear guidelines to parents and shared-care hospitals on how to treat ALL patients with fever, regardless of neutrophil count. Prediction scoring systems could supplement multidisciplinary efforts specifically involving front line emergency department staff to improve early access to antibiotics and supportive care as a whole. The only problem is that most shared-care hospitals in LMIC settings lack staff who can learn and implement such a prediction score system, given that children with ALL probably only form a minority of cases seen.

Protocol-specific analysis of infections during treatment may help inform positive changes in protocol design. In MS2003 [52], after 2 weeks of Protocol III DI, severe neutropenic fevers were observed. These observations led to the one-week mandatory break after Protocol IIIa, which helped reduced infective complications in the successor MS2010 protocol, unlike the AIEOP-BFM ALL 2000 experience.

## 16. Improving Supportive Care

Up to now, we have focused on adjusting various aspects of low-risk ALL therapy to the limited supportive care available in LMIC. Good supportive care is the bedrock of our improved cancer outcomes. Without good supportive care, most of what we propose is not possible.

Improving supportive care is cost-effective. Cost-effective measures like setting up appropriate inpatient and outpatient childhood cancer wards is transformational. Cohorting children with cancer who are immunocompromised in a childhood cancer unit reduces cross infections. Common childhood viral infections such as measles and varicella are mild in normal children but can be devastating in immunocompromised ones. With childhood cancer units, both doctors and nurses can be trained to implement life-saving neutropenic fever protocols immediately and give chemotherapy safely. Overcrowding is detrimental. Low-risk ALL can be treated in dedicated outpatient cancer centers where chemotherapy beds can be quickly cleaned and reused after a short IV vincristine or IM L-asp. By recycling outpatient beds, more children can be treated, reducing overcrowding and infections [63].

Setting up a childhood cancer unit must come with improved infectious disease and intensive care unit support. Laboratory tests including FBC and blood cultures, and a safe blood supply, are critical.

## 17. Overview of Maintenance Therapy

Maintenance therapy is indispensable in the cure of childhood ALL and is universally part of all chemotherapy protocols for ALL [64,65]. However, the exact reason for its essentiality remains unclear. Compared to the intensive prior phases of ALL therapy, MT is only mildly intensive and simple: it comprises a 2 drug “anti-metabolite” backbone of daily oral mercaptopurine (6-MP), and weekly oral/IV/IM MTX. From initial diagnosis, the duration of ALL therapy should exceed 2 years (104 weeks). Attempts to reduce the duration of ALL therapy to 12 months (TCCSG) to 18 months (BFM) have resulted in poorer outcomes. The only times that MT is omitted is after bone marrow transplantation or CAR-T cell therapy.

## 18. Duration of Maintenance Phase

Although MT is only mildly intensive, toxicities remain and even deaths occur [66]. During MT, patients remained mildly immunocompromised, exposing them to bacterial, fungal, and viral infections. The long duration of MT contributes to the cumulative risk of toxicity. Attempts to intensify MT by adding VCR/Dexa pulses and rotating drug pairs such as cyclophosphamide/cytarabine (SJ Total protocols), can add to the risk of infection (see below).

Historically, for ALL outcomes, boys fared worse compared to girls. However, with modern day risk-stratified therapy, this survival gap between boys and girls has narrowed [67]. In MS2003, which is Dexamethasone-based, boys and girls do equally well. Because of this, boys are not treated differently. However, due to inferior outcome in boys, some groups treated boys with an additional year of maintenance chemotherapy [67]. Thus, most contemporary protocols no longer treat them with separate durations, with very few exceptions (such as the TPOG trial group) [68].

Compared to standard duration of 2 years, a longer duration (i.e., 3 years) of MT confers no overall survival. Although longer MT reduces risk of relapse, it increases toxic deaths which erases the survival advantage [66]. Moreover, relapse on therapy is less salvageable. Numerous attempts have been made to shorten the duration of MT down to as little as 6 months, mostly with significantly poorer results [69,70,71]. However, certain small subsets of patients were cured despite shorter treatment. In the Tokyo Children’s Cancer Study Group’s L92-13 study, where MT was truncated to 6 months, patients with *TCF3-PBX1* or *ETV6-RUNX1* fusion had favorable survival [69]. Surprisingly, hyperdiploid ALL, which is low genetic risk group, fared the worst with shortened MT. However, these analyses are based on retrospective data of failed attempts to reduce duration of therapy. A shortened duration of MT generally resulted in a higher overall rate of late relapses. Therefore, 2-year ALL therapy including MT remains the current de facto strategy.

An intriguing model to reduce the risk of immunosuppression and infectious toxicity is the Brazilian Childhood Cooperative Group for ALL Treatment’s (GBTLI) use of an intermittent schedule of 6-MP and MTX [72]. Children were randomized to receive either continuous therapy (i.e., continuous oral 6-MP 50 mg/m^2^ daily and intramuscular MTX 25 mg/m^2^/week) or intermittent therapy (i.e., intermittent 6-MP 100 mg/m^2^ daily for 10 days and 11 days’ rest, plus MTX 200 mg/m^2^ as 6 h IV infusion every 3 weeks, with leucovorin rescue). Here, they found that children with LR ALL treated with the intermittent schedule had improved survival than those receiving the standard continuous schedule. Significantly, there was lower severe toxicity even though the overall cumulative MTX dose was higher in the intermittent group. Notably, boys allocated to the intermittent regimen had significantly better EFS than those receiving the continuous schedule.

## 19. VCR/Steroid Pulses

The addition of vincristine plus steroid (VCR/steroid) pulses during maintenance therapy significantly improved EFS by at least 10% in multiple clinical trials in the 1980s [66,73]. This was subsequently adopted by all major study groups. However, the benefits of these VCR/steroid pulses in contemporary more intensive protocols are increasingly questioned. The International BFM (I-BFM) Study Group prospective randomized multi-protocols study found that IR patients who received intensive ALL BFM-backbone protocols did not benefit from six pulses of VCR/Dexa during MT [74]. A recent large randomized trial from China showed that omitting these pulses in LR patients did not impact survival outcomes [75]. However, the EORTC ALL 58951-trial showed better survival [76]. With the contemporary intensive BFM-ALL protocol, LR or IR patients probably do not need VCR/steroid pulses during MT. However, for HR patients, the role of VCR/steroid pulses during MT is still unclear.

For the lower-risk groups, reduction of VCR/steroid pulses has been studied. The COG AALL0932 for NCI-SR B-ALL found that reducing the intensity of VCR/steroid pulses from 4 weekly to 12 weekly maintained the same excellent outcomes (OS 98%), although this was not performed in the context of a non-inferiority trial [77]. In the Ma-Spore trials, VCR/steroid pulses in the SR and IR arms were also given 10 weekly (MS2003) and 12 weekly (MS2010), with excellent outcomes [18,19]. Taken together, reduction or removal of these pulses might be applicable to those with the most favorable risk groups; i.e., favorable molecular subtypes with negative minimal residual disease (MRD) throughout [78]. MS2020 will continue VCR/Dexa pulses during MT every 12 weeks for SR/IR patients and 4 weeks for HR patients.

It is important to remember a successful ALL protocol is tested as one protocol with many phases. The intensity of all the phases contributes to the successful outcome. Currently, most contemporary protocols in HIC utilize sustained and highly intensive induction and reintensification blocks. This intensive ALL-BFM and augmented BFM backbone have been highly effective. In HIC, it is feasible for families to focus on an intensive 1 year of therapy and then a lower-intensity MT. With an intensive first year of therapy, the subsequent use of VCR/steroid pulses in MT is probably less important [73]. Whether a similar finding will result from a less intensive initial backbone, such as earlier trials that showed that VCR/Dexa pulses were useful, remains to be determined. In countries with limited resources for supportive care, a more spaced out and moderate intensity protocol during the first year followed by a slightly more intensive MT (with VCR/steroid pulses) might be more manageable. However, it is to be noted that these monthly VCR/Dexa pulses during maintenance can cause severe infections, especially with prolonged Dexa pulses of 2 weeks. VCR/Dexa pulses during MT may be complicated by varicella, measles, multi-resistant bacteria, and fungal infections. For low-risk ALL in LMIC, starting with less intensive upfront phases, shorter blocks of dexamethasone (6 mg/m^2^/day for 5–7 days without tailing) and one dose of VCR every 4-6 weeks, is recommended. To further mitigate risk of infections, some groups have even stopped MP/MTX during the weeks of VCR/Dexa pulses with no significant issues.

## 20. TPMT and NUDT15 Variants on 6-MP Metabolism

Mercaptopurine (6-MP), the main anti-metabolite medication used in MT, exhibits wide interpatient variability in its efficacy and toxicity. In dosing of 6-MP, the two actionable pharmacogenetic variants are TPMT and NUDT15. TPMT variants are common in Caucasians (10%) while NUDT15 variants occur more frequently in 20% of Asians.

TPMT methylates 6-MP and thioguanine, reducing their efficacy. Low TPMT activity increases the levels of active metabolites of thiopurines (TGNs), causing myelosuppression [79,80]. NUDT15 encodes a nucleoside diphosphatase which degrades thioguanine triphosphates by dephosphorylation. This dephosphorylation of thioGTP reduces its incorporation into DNA and protecting cells from apoptosis [81,82]. TPMT and NUDT15 variants have low enzymatic activities and these act in a co-dominant manner. Specifically, heterozygosity of TPMT and NUDT15 variants reduces the levels of TPMT and NUDT15 activities, causing mild sensitivity to 6-MP. Yang et al. proposed a thiopurine genetic score incorporating both TPMT and NUDT15 variants. In the MS2010 study [19], score 1 patients who carried either the TPMT or NUDT15 variant tolerated a reduced dose of 6-MP at 40 mg/m^2^/day. The frequency and type of variants affecting both enzymes vary significantly by ethnicity [83,84].

Pre-emptive testing of both TPMT and NUDT15 for possible dose modification is now standard care, with carefully established guidelines [85]. This is because doses that are customized based on TPMT or NUDT15 status reduce the likelihood of acute and severe toxicities (e.g., myelosuppression), without compromising disease control. Therefore, the risk-benefit ratio of pre-emptive genotyping is favorable and should be implemented in regions likely to have a high allelic frequency of these variants, and where testing resources are available.

## 21. Conclusions

Taken together, NCI SR features, low-risk genetics (hyperdiploidy, ETV6-RUNX1) and a rapid early response identify a group of patients who can be cured with low-intensity ALL therapy. Even in LMIC settings, these low-risk patients can be identified and cured cost-effectively with low-intensity protocols. Low-intensity protocols are based on two principles of (1) starting slow, with a 3-drug, anthracycline-free induction, delaying first IT to day 8, and (2) keeping safe, with low-intensity DI and uninterrupted metronomic MP/MTX maintenance. Setting up the appropriate supportive care to support the treatment protocol is as important. Adapting and testing therapy appropriate to resource-constrained supportive care and testing for TPMT/NUDT15 variants in high-frequency areas can be cost-effective. To appropriately adapt the best standards of care, partnering aspirant institutions through St Jude Global and SIOP is key. As a community caring for children with cancer, we have been fortunate. Realizing that childhood cancer is rare and we cannot do it alone make sharing experience and working together the guiding principles. By learning how to better treat low-risk ALL cost-effectively, LMIC could potentially contribute to the global ALL knowledge of how to cure with less. We are hopeful that HIC, in the near future, can learn from LMIC on “curing the curable” with less. As teachers, we learn best from our students.

## Figures and Tables

**Figure 1 jcm-10-04728-f001:**
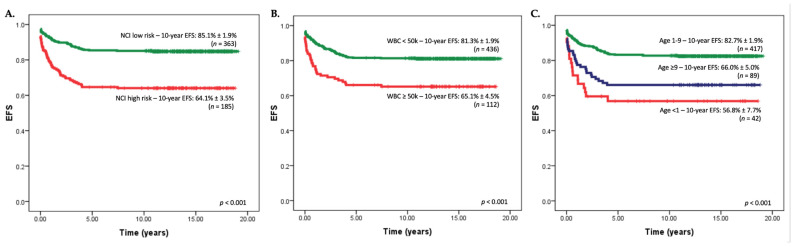
(**A**–**C**) Kaplan-Meier 10-year event-free-survival (EFS) curves from the Malaysia-Singapore ALL 2003 study cohort: significant differences in outcomes based on features from the National Cancer Institute (NCI) risk criteria such as total white blood cell (WBC) count at diagnosis and age were found; in this study risk stratification was based primarily on end-of-induction PCR-MRD responses. (**A**) 10-year EFS based on the presence of NCI low versus high-risk criteria, (**B**) 10-year EFS based on total white cell count at diagnosis, (**C**) 10-year EFS based on NCI criteria defined age groups. NCI risk criteria has been shown to clearly define groups of patients with clinically significant differences in long term EFS regardless of MRD response at the end of induction. This is a finding that is especially relevant to children in LMIC settings who may not have access to MRD monitoring and risk stratification during treatment. Children with lower risk ALL can already be defined early at the point of diagnosis based on NCI risk features.

**Figure 2 jcm-10-04728-f002:**
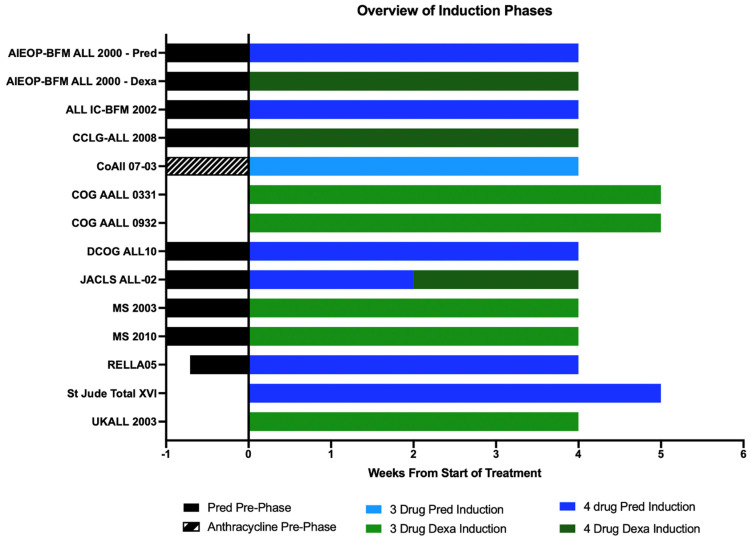
Induction regimens in contemporary ALL studies for low−risk patients. Prednisolone (Pred) prophase allows management of tumor lysis. 3 drug induction refers to induction protocols which include the use of Prednisolone or Dexamethasone (Dexa), Vincristine and L-asp, with the exception of patients treated on the CoALL 07-03 [21] who received 3 drug Prednisolone-based induction consisting of Prednisolone, Vinristine, and either Doxorubicin or Daunorubicin, without L-asp. This included a pre-phase comparing the responses after a single dose of either Doxorubicin or Daunorubicin. 4 drug induction protocols include the use of anthracyclines such as Doxorubicin or Daunorubicin. For LMIC, 3-drug Dexa-based induction is safer. For 4-drug induction, Pred-based induction is probably less toxic than one that is Dexa-based.

**Figure 3 jcm-10-04728-f003:**
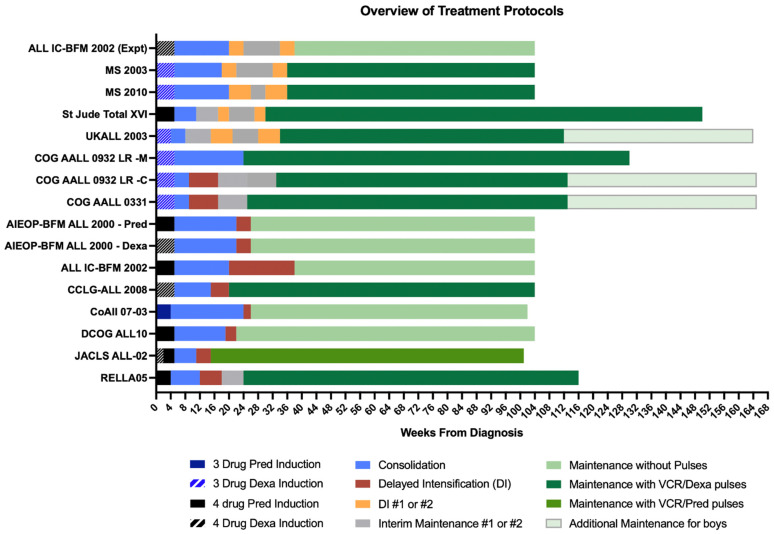
Overview of contemporary ALL protocols for lower-risk ALL: Major differences in protocol design of delayed intensification (single versus double blocks interspersed with interim maintenance blocks), types of maintenance phases are highlighted. Consolidation phases described in this figure refer to the period following the completion of induction phase and end prior to the start of delayed intensification phase and therefore include HD MTX phases. The experimental arm of the **ALL IC-BFM 2002 study** [16] comprised two shorter DI blocks (Protocol III), split from the original single Protocol II DI. **MS2003** [18] and **MS2010** [19] studies employed a similar dosing strategy with multiple DI blocks with improvements in toxicity following dosing modifications. The **St Jude Total XVI** [26] protocol embeds dual 3 week blocks of DI in interim maintenance phases which start right after the consolidation phase. Similarly, the **UKALL 2003** [27] protocol also comprises two blocks of DI, albeit longer in duration and higher in dose intensity as compared to the abovementioned studies. **COG AALL 0932** [22] randomized low-risk patients to receive either P9904 regimen A-based (Arm LR-M), which is a very low-intensity protocol without alkylating agents or anthracyclines, or the CCG 1991 regimen-like outpatient-based regimen (Arm LR-C) with reduced vincristine/dexamethasone pulses during maintenance phase (every 12 weeks). For patients assigned to Arm LR-M, the total duration of therapy would be 2½ years from diagnosis for both female and male patients. For those assigned to Arm LR-C, the duration of therapy would continue to be gender based: 2 years from the start of interim maintenance for female patients and 3 years from the start of Interim Maintenance I for male patients. In the **COG AALL 0331** [23] study, patients in the lowest defined risk group in the study, the standard risk-low group, were randomized to receive either standard treatment or four additional doses of PEG L-asp at 3 week intervals in an attempt to intensify treatment to improve outcomes in this group of patients. Although intensification failed to improve outcomes, the authors concluded that standard COG therapy without intensification still led to excellent outcomes in this identified low-risk group. **AIEOP-BFM ALL 2000** [36] randomized patients to receive either Prednisolone or Dexamethasone during induction. The **CCLG-ALL 2008** [17] study was based on BFM ALL treatment backbone but modified to reduce toxicity in SR patients by halving the dose intensity of early intensification after induction and before consolidation. DI was modified as per the COG with 25–33% reduction of Dexamethasone and Doxorubicin. Patients were then randomized in the maintenance phase to either receive standard maintenance therapy with vincristine and dexamethasone pulses versus a 1 week rest of mercaptopurine and MTX during the vincristine dexamethasone pulse. The lowest-risk group of patients treated on the **CoALL 07-03** [21] trial was given the reduced intensity LR-R arm with only 1 week of Dexamethasone, two doses of Vincristine and single doses of Doxorubicin with PEG L-asp in a shortened DI protocol. **DCOG ALL10** [24] includes a significantly deintensified DI Protocol IV with only 2 weeks of Dexamethasone, two doses of Vincristine and a single dose of PEG L-asp; this was followed with maintenance therapy consisting only of oral 6-MP and MTX without any pulses. The **JACLS ALL-02** [25] protocol used Prednisolone pulses during maintenance in contrast to most other groups where Dexamethasone was used during pulses with Vincristine during the maintenance phase; Pirarubicin was also used instead of the more commonly used Doxorubicin or Daunorubicin as the anthracycline of choice during induction and DI.

**Figure 4 jcm-10-04728-f004:**
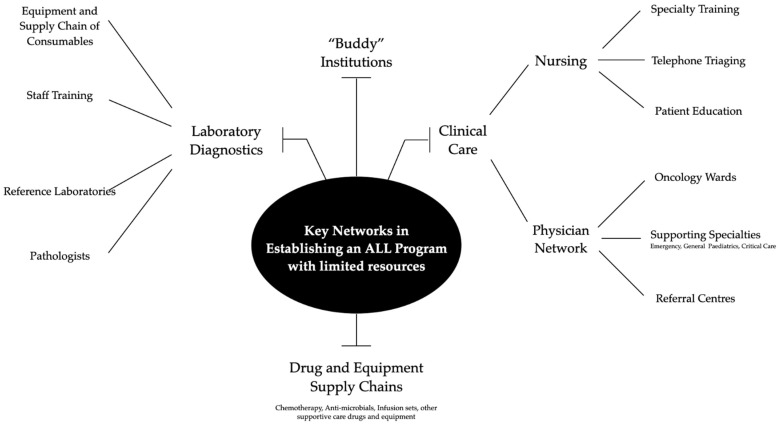
Proposed key networks in establishing an ALL program with limited resources. “Buddy” institutions refer to more established programs that newer growing programs with limited resources can reach out to for help and advice regarding patient care and technical support.

**Table 1 jcm-10-04728-t001:** Key stratified strategies discussed in this review on management of low-risk childhood ALL.

	Low-Income Countries (LIC) Setting	Middle-Income Countries (MIC) Setting	High-Income Countries (HIC) Setting
Low-Risk features	NCI SR CNS I No mediastinal mass	NCI SR CNS I Flow T vs. B DNA index Cytogenetics/FISH OFT screening	NCI SR CNS I Flow T vs. B Cytogenetics/FISH OFT screening/RNA-Seq
Early response PB	D8 blast < 1 × 10^9^/L	Day 8 blast < 1 × 10^9^/L	Day 8 blast or PB flow MRD
Early response BM	Day 15 M1/2, EOI M3	Day 15 M1/2, EOI M3 Flow-MRD-lite	Flow MRD or PCR MRD at EOI, EOC
Protocol	One protocol	SR/HR	SR/IR/HR
B-ALL SR	VCR-Dexa	3-drug Dexa-based	3-drug Dexa-based
T-ALL and B-HR	VCR-Dexa	3-drug Dexa-based	4-drug Pred-based
Central Nervous System (CNS) directed Rx	Cranial RT/IT MTX	IT MTX/Cranial RT	IT MTX/HDMTX
Delayed intensification	VCR-Dex-Doxo	VCR-Dex-L-asp + CTX-araC-MP	Protocol II
Maintenance Therapy	4-weekly VCR/Dex pulse	8-weekly VCR/Dex pulse	12-weekly VCR/Dex pulse
TPMT and NUDT15	In sensitive patients	In sensitive patients or routinely	Routine
Clinical Trial Examples	LMIC Examples: RELLA05 [14] ICiCLe ALL 14 [15]	ALL IC-BFM 2002 [16] CCLG-ALL 2008 [17] MS 2003 [18]/ 2010 [19]	AIEOP-BFM ALL 2000 [20] CoALL 07-03 [21] COG AALL 0932 [22]/0331 [23] DCOG ALL10 [24] JACLS ALL-02 [25] St Jude Total XVI [26] UKALL 2003 [27]

**Table 2 jcm-10-04728-t002:** Risk stratification criteria, cumulative chemotherapy dosing and proportions of patients treated in lowest-risk arms.

	B-ALL	T-ALL	Age Criteria	Total White Cell Count at Diagnosis	Central Nervous System (CNS) Status	Specific Cytogenetic Inclusion Criteria	Prednisolone Response	Treatment Responses				Cumulative Cyclophosphamide Dosing (mg/m^2^)	Cumulative Anthracycline Dosing (mg/m^2^)	Cumulative L-Asparaginase × 1000 units/m^2^ Dosing ^^	% of Study Population Treated in the Lowest-Risk Arms
								D8	D15	End of Induction	Later Timepoints				
***AIEOP-BFM ALL 2000* [20,36]**	Y	Y	1–17	Any	Any		Y	-	-	PCR MRD NEG	Day 78 PCR MRD NEG	3000	240	80	28
***ALL IC-BFM 2002* [16]**	Y	Y	1–6	<20,000/μL	Any		Y	-	M1/M2	-	-	3000	180	Standard: 80 Expt: 120	31
***CCLG-ALL 2008* [17]**	Y	-	1–10	<50,000/μL	No CNS 3		Y	-	M1/M2	PCR MRD NEG or Flow MRD < 0.01%	Week 12 PCR MRD NEG or Flow MRD < 0.01%	2000	125	80	39
***CoAll 07-03* [21] *LR-R***	Y	-	<10	<25,000/μL	Any		-	-	MRD < 0.01%	^⍬^ MRD < 0.01%	-	900	120	125	13
***COG AALL 0331* [23,37] *LRS***	Y	-	1.01–9.99	<50,000/μL	CNS 1 only	Triple trisomies of chromosomes 4, 10, and 17 Or ETV6-RUNX1	-	M1 *	M1 *	^⍬^ MRD < 0.01%	-	1000	75	80	35
***COG AALL 0932* [22] *LR-C***	Y	-	1.01–9.99	<50,000/μL	CNS 1 only	Triple trisomies of chromosomes 4 and 10 Or ETV6-RUNX1	-	PB MRD < 0.01%	-	^⍬^ MRD < 0.01%	-	1000	75	80	3.5
***COG AALL 0932* [22] *LR-M***	Y	-	1.01–9.99	<50,000/μL	CNS 1 only	Triple trisomies of chromosomes 4 and 10 Or ETV6-RUNX1	-	PB MRD < 0.01%	-	^⍬^ MRD < 0.01%	-	0	0	40	3.5
***DCOG 10* [24]**	Y	Y	1–18	Any	CNS 1 only		Y	-	-	PCR MRD NEG	Day 80 PCR MRD NEG	2000	120	80	26
***JACLS-ALL-02* [25]**	Y	-	1–9	<10,000/μL	No CNS 3		Y	-	M1/M2	M1	-	1500	^ 90	72	40
***MS 2003* [18]**	Y	Y	0–18	Any	CNS 1 only		Y	-	-	PCR MRD NEG	Week 8 PCR MRD NEG	3000	120	140	31
***MS 2010* [19]**	Y	-	1–10	Any	CNS 1 only		Y	-	-	PCR MRD NEG	Week 8 PCR MRD NEG	3000	0	167.5	40
***RELLA05* [14]**	Y	-	1–9	<50,000/μL	CNS1/CNS2 Traumatic LP	DNA index of ≥1.16 or ETV6-RUNX1	-	-	** MRD <0.01%	-	-	0	50	160	22
***St Jude Total XVI* [26]**	Y	-	1–10	<50,000/μL	Any	DNA index of >/=1.16 or ETV6-RUNX1	-	-	MRD < 1%	-	D46 MRD <0.01%	1000	110	208	43
***UKALL 2003* [27]**	Y	Y	1–10	<50,000/μL	Any		-	-	M1/M2	+ MRD < 0.01%	-	2000	150	64	33

* D8 or 15 M1 marrow; ** Day 19 MRD < 0.01%; + D29 MRD detectable but <0.01% AND undetectable MRD before start of interim maintenance; ^⍬^ Day 29 responses; ^ Pirarubicin; ^^ Where pegylated L-asp was used, 2500 units/m^2^ was calculated to be the equivalent of 40,000 units/m^2^ given over 1 week; DNA Index: (the ratio of DNA content in leukemic cells to that in normal diploid G0/G1 cells).

**Table 3 jcm-10-04728-t003:** Consolidation and MTX dosing across clinical trials.

	Cumulative Int./High Dose MTX Dose (g/m^2^)	Number of Int./High MTX Doses	Dose of Int./High Dose MTX (g/m^2^)	Duration of 6MP (weeks)	Daily 6MP Dose (mg/m^2^)	Total 6MP Dose (mg/m^2^)	Number of Intrathecal Chemotherapy Injections	Other Drugs
***AIEOP-BFM ALL 2000* [20,36]**	20	4	5	8	25	1400	4	-
***ALL IC-BFM 2002* [16]**	8	4	2	8	25	1400	4	-
***CCLG-ALL 2008* [17]**	8	4	2	8	25	1400	4	-
***CoAll 07-03* [21]**	3	3	1	2	100	1400	3	Teniposide 165 mg/m^2^ + Thioguanine (100 mg/m^2^/day) for 1 week + L-asp 45,000 units/m^2^ + PEG-Asp 5000 units/m^2^ + Cytarabine 12,300 mg/m^2^
***COG AALL 0932* [22] ***(LR-M)*****	6	6	1	19	50	6650	6	Dexamethasone 84 mg/m^2^ + Vincristine 6 mg/m^2^
***DCOG 10* [24]**	20	4	5	8	25	1400	4	-
***JACLS-ALL–02* [25] *Arm A***	6	2	3	1	50	350	4	Cyclophosphamide 1.5 g/m^2^ + Cytarabine 750 mg/m^2^
***JACLS-ALL–02* [25] *Arm B***	6	2	3	-	-	-	4	Dexamethasone 50 mg/m^2^ + Cyclophosphamide 1 g/m^2^ + Cytarabine 500 mg/m^2^
***MS 2003* [18]**	8	4	2	8	25	1400	4	-
***MS 2010* [19]**	10	4	2.5	8	25	1400	4	Interspersed Cyclophosphamide blocks
***RELLA05* [14]**	10	4	2.5	8	50	2800	4	-
***St Jude Total XVI* [26]**	10	4	2.5	8	50	2800	4	-
	**Total Dose of Dose Escalating MTX (g/m^2^)**	**No. of Dose Escalating MTX**	**Oral MTX (mg/m^2^)**	**Duration of 6MP (weeks)**	**Daily 6MP Dose (mg/m^2^)**	**Total 6MP Dose (mg/m^2^)**	**Number of IT Chemotherapy Injections**	**Other Drugs**
***COG AALL 0932* [22] *(LR-C)***	1	5	-	4	75	2100	4	Vincristine 9 mg/m^2^
***COG AALL 0331* [23,37]**	1	5	-	4	75	2100	4	Vincristine 9 mg/m^2^ (L-asp intensification arm: 4 additional doses of PEG-Asp (10,000 units/m^2^)
***UKALL 2003* [27]**	-	-	140	4	75	2100	4	Vincristine 4.5 mg/m^2^

Differences in MTX dosing strategies are summarized and highlighted in this table, from intermediate to high-dose MTX regimens to low-dose MTX regimens including the characteristic COG dose-escalating MTX. Patients on the **CoALL 07-03** [21] trial were treated with intermediate doses of MTX but were given a combination of other drugs such as Teniposide, L-asp, and Cytarabine as well. In the **COG AALL 0932** [22], low-risk patients were randomized to receive either the P9904 regimen A-based (Arm LR-M) which included 6 courses of intermediate dose (1g/m^2^) MTX without any further alkylating agents or anthracyclines, essentially omitting DI entirely and completing therapy with the maintenance phase; or the CCG 1991 regimen-like outpatient-based regimen (Arm LR-C) with standard COG dose-escalating MTX. Patients treated on the **JACLS ALL-02** [25] were randomized to receive either truncated BFM-like consolidation (Arm A) or low-dose Cytarabine containing consolidation (Arm B). Of all the reviewed studies, the **COG AALL 0932** [22] and **COG AALL 0331** [23] studies, together with the **UKALL 2003** [27], included the lowest intensity of MTX treatment. While of interest to LMIC or LIC groups with limited access to serum MTX drug monitoring, the excellent outcomes achieved in these studies were derived from HIC settings with individual protocol-specific caveats such as the more stringent criteria imposed by the COG to be considered as low risk, and later intensification in other parts of the protocol in the UKALL 2003, which have been reported to be toxic even in HIC settings.

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
