# Peer review of "Curing the Curable: Managing Low-Risk Acute Lymphoblastic Leukemia in Resource Limited Countries"

_jcm, 2021, doi:10.3390/jcm10204728_

Round 1

Reviewer 1 Report

The overview is an excellent summary of current pediatric ALL treatment schemes. It focuses on low-risk groups and discusses in each chapter the potential low risk treatments adaptable to LMIC.

It is not proposing a therapeutic regimen for the setting of LMIC but showing the directions where such a protocol might be developed from. Therefore, it is a valuable summery with high clinical and practical impact.

Major points:

- Line 92 - Chapter 4: The importance of the NCI SR criteria: I would assume that the NCI criteria are well known to hemato-oncologists, so this chapter might be somewhat shortened and rewritten in a more concentrated manner. Table 2 is very helpful in this aspect - and especially the last column is informative: Of what use is a SR group comprising only 3.5% of the patients?

- Line 158: Referencing and partnering is a critical process in pediatric oncology, not only for technical reasons, as you state here. I think this would be one of the central themes in establishing ALL-treatment options in hospitals that didn't do it before, as the authors describe so nicely with the Ma-Spore group in chapter 14: The treating pediatrician or general physician needs this network: Needs to know where to call for questions and medical and technical support. I think this subject is so central that you extend this chapter after describing the Ma-Spore experience and generalize it: And as I would see it: The technical aspect is only one side: The same importance has training and counselling.

- Line 557: Conclusion. This chapter is very short. It may be of help when it would contain the major conclusions out of the various chapters. This may help the reader to be able to imagine the frame for a potential future treatment protocol in LMIC.

- Table 1: Very good overview. You should add references for exemplary trials. And LIC, MIC and HIC sould be introduced (not only NMIC).

Minor points:

Line 23. You may close the last phrase like: “…is to cure low-risk ALL with less intensity and less toxicity” or similar because this is the way to achieve your goals as you write.

Lines 44 to 53: Please insert quotes that support the content.

Line 79. I suppose it is an optimistic expectation for hospitals to set up their own PCR-fusion testing without external training. 

Line 195: The chapter on T-ALL might be focused more on SR T-ALL omitting the - surely interesting - results on IR and HR T-ALL.

Table 3: Please give the citations for the indicated protocols.

Author Response

请参阅附件

Reviewer 2 Report

Dear Authors

The review "Curing the curable: Managing Low-Risk Acute Lymphoblastic Leukemia in Resource Limited Country" is very interesting and draws attention to differences in treatment outcomes between countries.

I have two questions - 1. Have Authors met other examples of cooperation between countries than Ma-Spore Sudy Group?

2. How to improve supportive care in LMIC?

Reviewer 3 Report

To the authors:

This review aims to describe the possible treatments for childhood acute lymphoblastic leukemia (ALL) in low-middle income countries. Moreover, they reviewed the treatment components of  low-toxicity regimens in recent clinical trials for low-risk ALL and suggest how they can be adopted in LMIC. I would like to congrats the authors for this review, because I believe that cancer treatments should not be used only in high-income countries and we should join efforts so that more proposals such as those presented in this review are carried out. Nevertheless, and in spite of the significant amount of work performed, some important issues have to be consider in order to accept this review.

  1. The review lacks bibliographic references. The authors use only 6 bibliographic references from the Introduction to section 3 (lines 27-91). There are complete paragraphs without any reference. This could be defined as plagiarism if the authors do not add the references in each sentence that they have used from a study carried out by another group. For instance, NCI standard-risk has no reference. Did the authors invent the NCI criteria or the NCI standard-risk?
  2. In the same way, in Table 1 and Table 2 the authors should add the references of the studies from which they have obtained the data described in the tables. They should add a new column to include the references of the studies used.
  3. The authors the authors make excessive use of acronyms, which in many cases do not define their meaning (WBC, SIOP, WHO, CNS, BFM…). Some parts of the review are hard to understand by the readers. They should reduce the use of them and write the complete term when they use the first time an acronym.
  4. In table 1, the authors describe HIC and HIC-trial variables, but what is the difference between the HIC and HIC-trial columns?
  5. The authors should add figure legends with more in detail information. For instance, in figure 2, authors used 3-drug Dexa induction and 4 drug Dexa induction, but what do these terms mean?
  6. The nomenclature used to define patients, protocols, research groups and studies is confused. I recommend establishing a simple code to facilitate the identification of each one in a clearer and simpler way for the reader.

Overall, the review is interesting but contains too much information. The tables and figures should be explained in more detail to facilitate understanding of the data shown.

Author Response

请参阅附件

Reviewer 4 Report

The manuscript is well written and is a comprehensive description of conventional chemotherapy treatment of childhood ALL with a focus on the experience in Singapore and Malaysia and implications for the management of ALL in resource limited Countries. The title is misleading in my opinion. The manuscript in fact offers suggestions for treatment of all patients, not only the “curable ones”. I would suggest something like: “Childhood ALL treatment strategies worldwide and the Malaysia - Singapore experience: indications for Countries with limited resources.”

The manuscript should include a short chapter on the definitions of LIC, LMIC, HMIC and HIC, and explain that even within these categories there is a lot of variability, in part due also to the health system and social organization. Suggestions here provided in my opinion apply mostly to Middle Income Countries. Results obtained in HIC with low intensity protocols in general cannot be reproduced in Countries with limited resources; the successful experience of RELLA study was conducted in a structure comparable to those of HIC. Childhood ALL treatment must thus be adapted to local specificities.

I have only minor editorial suggestions and comments, as listed below.

ALL-IC-2002 should be “ALL IC-BFM 2002”, please check the whole text.

Lines 96-102: RNA seq is generally not available in LIC or LMIC and only occasionally in HMIC

Line 131: it should be “age ≥10” instead of “age >10”

Table 2: name of the study protocols: AIEOP-BFM ALL 2000; ALL IC-BFM 2002; CoALL

Legend Figure 2. “and allow confirmation of diagnosis of ALL” is unclear and not needed in my opinion.

Line 244: should be “one third reduction of the risk of relapse” instead of “one third the risk of relapse”

Line 281: both Leunase and Kidrolase are derived from Kiowa Hakko

Line 287: “who” should be “with”

Table 3: Time infusion of HDMTX may be added. AIEOP-BFM ALL 2000: phase duration is 8 weeks (not 17), should be the same for ALL-IC BFM and DCOG; please double check for all Groups

Line 350: Protocol IV: the phrase is confounding, it should be clear that Protocol IV consists of Dexamethasone, Vincristine 2 doses, Peg-L-Asp (one dose) and I.T. therapy (one dose).

Line 438: need for some editing

Line 449: MTX SC should be IM

Line 469: “survival” should be “survival improvement”

Line 498: it should be specified that each of the 6 pulses included two doses of vincristine

Line 534 – 549 need minor editing to correct some errors

Author Response

请参阅附件

Round 2

Reviewer 1 Report

Thank you for revising the manuscript and anwering the topics in question.

Author Response

Thank you for your affirmation of our work.

Reviewer 3 Report

Comments to the Author:

I commend the effort made by the authors to revise this manuscript. The authors have answered correctly all my questions and they have added the references in the proper manner. They have improved the tables 1 and 2 into the study, as I suggested to them, and Table 1 is now much more understandable and clearer. The authors have included the references into the tables too. I consider that now the manuscript is more solid and it has improved considerably for its publication. However, I still have an important issue that should be addressed:

  1. There is not Table 1 Legend. It should be added.

Author Response

Thank you for your further review, we have added Table 1 Legend.